# Plant Natural Products: Promising Resources for Cancer Chemoprevention

**DOI:** 10.3390/molecules26040933

**Published:** 2021-02-10

**Authors:** Li Ma, Mengmeng Zhang, Rong Zhao, Dan Wang, Yuerong Ma, Li Ai

**Affiliations:** 1School of Basic Medicine, Chengdu University of Traditional Chinese Medicine, Chengdu 611137, China; mlhr940303@163.com; 2School of Pharmacy, Chengdu University of Traditional Chinese Medicine, Chengdu 611137, China; garita119@163.com (M.Z.); 15228279571@163.com (R.Z.); q1833374665@163.com (D.W.); 3School of Ethnic Medicine, Chengdu University of Traditional Chinese Medicine, Chengdu 611137, China

**Keywords:** plant, natural product, cancer chemoprevention

## Abstract

Cancer is a major factor threatening human health and life safety, and there is a lack of safe and effective therapeutic drugs. Intervention and prevention in premalignant process are effective ways to reverse carcinogenesis and prevent cancer from occurring. Plant natural products are rich in sources and are a promising source for cancer chemoprevention. This article reviews the chemopreventive effects of natural products, especially focused on polyphenols, flavonoids, monoterpene and triterpenoids, sulfur compounds, and cellulose. Meanwhile, the main mechanisms include induction of apoptosis, antiproliferation and inhibition of metastasis are briefly summarized. In conclusion, this article provides evidence for natural products remaining a prominent source of cancer chemoprevention.

## 1. Introduction

With rapid development of population growth and lifestyle, cancer as a growing threat in the world, is the second leading noncommunicable disease of death globally next only to cardiovascular disease in the developed nations and even in many other developing countries such as China and India [1].The International Agency for Research on Cancer (IARC), a part of the World Health Organization (WHO), issued an acclivity of 19.30 million new cancer cases and 10.00 million cancer deaths in 2020. Among these 36 cancer types, breast cancer is the most generally diagnosed neoplasm, immediately followed by lung cancer, colorectal cancer, and prostate cancer, with incidences of 11.7%, 11.4%, 10.0%, and 7.3%, respectively. Over the same period, a tool that predicted an acclivity of new cancer cases from 19.30 million to 28.40 million in 2020 up until 2040 [2]. Actually, there is an urgent need to develop the mechanism-based approaches/strategies for the management of cancer, which can eliminate damaged or malignant cells.

At least 40% of alternative therapies such as herbal medicine were applied for preventing disease in the United States. Historically medicinal botanicals, as part of complementary medicine in the USA, have provided a source of inspiration for discovering novel drug agents, and more than half of currently available drugs are natural products or are related to them [3]. Of particular interest, considerable epidemiological evidence has been accumulated indicating that over 50% of the approved anticancer agents like retinoids and podophyllotoxin are either natural compounds or natural product derivatives derived from herbal medicine [4,5]. Furthermore, in the past two decades, numerous dietary and botanical natural products with the capability to regulate physiological functions like phenolics, flavonoids, alkaloids, carotenoids, gingerols, as well as organosulfur compounds, have been documented to suppress early and late stages of carcinogenesis in various in vivo and in vitro models [6]. Accordingly, there is a considerable scientific interest in the continuing discovery of effective anticancer agents from natural product sources.

Recently, cancer chemoprevention has been developed as a topic field of alternative/additional regimen, whereby administration of natural or synthetic agents for reducing, stabilizing, and reversing precancerous lesions. The current review is focused on the principles of cancer chemoprevention, potential active natural products in cancer chemoprevention and its related mechanisms of action, as well as introducing the future prospects for cancer prevention.

## 2. Methods

Relevant references were retrieved from PubMed, Web of Science, Springer, and Wiley Online Library databases from 1999 to 2019, and the search term included anticancer/tumor combined with natural products. Both in vivo and in vitro studies were involved in this review. The references of these articles were further checked to find more new studies. The search strategy identified 1586 publications, and 447 references were excluded for duplication. The remaining studies were further evaluated by the authors of first author (M.L., Z.M.M.), and then, two authors (Z.R., W.D.), and natural products with cancer preventive effects are mainly divided into five types of ingredients for introduction.

## 3. An Overview of Carcinogenesis

Carcinogenesis is a multistep process characterized by a progression of molecular changes that ultimately transform a cell to undergo uncontrolled proliferation. Human cancer is a progressive disease that develops in distinct stages by long periods of latency. Cancer development requires 20–40 years or more, and therefore, the initiation stage might occur decades before an individual is clinically diagnosed for cancer. Cells undergo changes with each disruption fundamentally featured by tumor initiation, promotion, and progression.

The occurrence of probable mutations within genes that encode growth factors, transcription factors, protein kinases, apoptotic signaling proteins, or adhesion molecules, predispose a cell to become tumorigenic during initiation. The additional irreversible genetic changes, accompanying with each new cycle of proliferation, were accumulated as “initiated cells”. Subsequently, the second stage, promotion, a relatively lengthy and reversible process of carcinogenesis, ultimately resulted from genetic and epigenetic alterations by repetitive cycles of proliferation, selective clonal expansion. The cells during promotion have acquired the ability to evade programmed apoptosis and immune surveillance, accompanying with sustained angiogenesis. Many cancers are not diagnosed until third stage, progression, in which the tumor becomes malignant and maybe metastasizes to multiple body regions. The tumors in this stage are very difficult to treat, which usually requires surgery in combination with chemotherapeutic drugs and radiotherapy.

Each individual has a unique cancer risk profile which is determined by sequential accumulation genetic and epigenetic factors over a period of many years. According to the carcinogenesis, cancer control strategies can be divided into three stages, including prevent/inhibit one third of cancers caused by tobacco and alcohol, cure/treat another third, and provide good, palliative care to the remaining third [1,7]. As further data accumulated, the transition from one cancer phase to the next can be stimulated by certain carcinogenic factors, such as obesity, alcohol, smoke, biological agents, infections, and ultraviolet radiation as well as an unhealthy dietary habit, and others [8], which ultimately resulted in invasive/metastatic carcinoma. It is predicted that between 30% and 50% of cancer incidences can be prevented with proper medical awareness [6]. Evidence indicated that two thirds of the cancer deaths in the US each year can be attributed to diet, physical activity habits, and cigarette smoking. Therefore, chemoprevention as a rapid evolving field, can perturb a variety of steps in tumor development to reduce or delay the occurrence of malignancy.

## 4. Scientific Principles of Cancer Chemoprevention

Interest in cancer chemoprevention of research has markedly increased with improved illustration of the biology of carcinogenesis and identification of potential molecular targets involving this process. It has become apparent that chemoprevention should incorporate the concept of ‘delay’, which implies that many years or decades may be added to human lifespan. This interest has been further aroused by successes in the prevention of prostate, breast, and colon cancers, and there are 10 medications that were approved by Food and Drug Administration (FDA) for the risk reduction of cancer [9].

Chemoprophylaxis, the term coined by Lee Wattenberg in 1966, is an effective method focused on inhibition of tumor/cancer through synthetic or naturally occurring agents [10]. Later, in 1976, Michael B. Sporn and other coauthors introduced the term “cancer chemoprevention” which was defined as the prevention of the cancer occurrence by administration of natural compounds [11,12]. Gary Kello, chief of the Chemoprevention Branch, extended the definition of “cancer chemoprevention” as utilization of agents to suppress, prevent, or reverse the carcinogenic process to invasive cancer [13]. Furthermore, other important considerations, “combination chemoprevention”, introduced in 1980 as a new strategy, can not only enhance potential synergistic efficacy of drugs, but also decrease the toxicity of the individual agents, with a lower dose treatment in a combination regimen. This strategy was clinically believed in a definitive investigation, demonstrating that the combination of difluoromethylornithine and sulindac showed more effective in preventing colon cancer [14]. Recently, chemoprevention was considered as an important and hopeful tool for controlling cancer. It plays a role in preventing the development of invasive and metastatic properties in established neoplasms, and this approach is to lower the rate of cancer incidence. Chemoprevention can be organized into three parts: (1) primary prevention, inhibiting the tumor in healthy individuals with high risk, for instance, hepatitis B vaccine; (2) secondary prevention, inhibiting tumor development in individuals with precancerous lesions like invasion; (3) tertiary prevention, inhibiting recurrence or second primary cancers in target patients [15].

## 5. Agents for Cancer Chemoprevention

### 5.1. Synthetic Drugs

Until now, an increasing number of agents have been used for preventing cancer in various experimental models, and many of these drugs have even been clinically proven. Among these drugs, the selective estrogen receptor modulators (SERMs), such as tamoxifen, raloxifene, and arzoxifene, are the most important class, which are also beneficial for preventing osteoporosis in addition to breast cancer prevention in women at risk [16,17,18].

Other significant chemopreventive agents in various stages of clinical trials include the androgen analogs (finasteride and dutasteride), aromatase inhibitors, anastrozole and exemestane, as well as aspirin and metformin. More interesting, metformin is recently being evaluated prospectively for many cancers [19,20]. These synthetic drugs in reality have been “repurposed” because they were used for many years in other disease treatments, rather than originally developed. This largely solved many issues of dosage selection and long-term safety before these agents entered into a clinical trial phase [21].

### 5.2. Natural Products

Besides, natural products derived from food, spices, and herbal medicine have gradually become the major resource for cancer chemoprevention. In contrast to many synthetic drugs, they have been extremely safe for human administration for thousands of years (Table 1).

#### 5.2.1. Polyphenols

Phenolic compounds derived from the main class of secondary metabolites, widely present in food and nutraceuticals, are natural phytochemicals mostly derived from phenylalanine and less from tyrosine [22]. Recently, increasing evidence in the scientific literature indicated that phenolic compounds possess an effective inhibitory effect on cancer invasion and metastasis.

Catechins polyphenols, including (−)-epicatechin (EC), (−)-Epigallocatachin (EGC), (−)-epicatechin-3-gallate (ECG), and (−)-epigallocatechin-3-gallate (EGCG), are the most abundant and representative compounds present in tea (*Camellia sinensis*, Theaceae) [23,24]. In particular, EGCG accounting for 50–80% of the catechin content, identified as one of the most effective compounds as an epigenetic modifier for cancer treatment and chemoprevention, has received recent attention. Furthermore, related experiments showed that both EGCG (0.01 and 0.1%) could inhibit the growth inhibition of the VEGF-VEGFR axis on hepatocellular carcinoma (HCC) cells [25]. In clinical trials, breast cancer patients treated with long-term EGCG plus radiotherapy for 2–8 weeks showed significantly lower serum levels of vascular endothelial growth factor (VEGF) and hepatocyte growth factor (HGF), as well as reduced activation of metalloproteinase-9 and metalloproteinase-2 (MMP9/MMP2), which are considered promising chemoprophylaxis targets. In addition, the serum of the patient not only inhibited the proliferation and invasion of the highly metastatic human MDA-MB-231 breast cancer cells in vitro, causing cell cycle arrest in G0/G1 phase, but also decreased the activation of MMP9/MMP2, Bcl-2/Bax, c-Met receptor, NF-κB expression, and Akt phosphorylation level. Apoptosis induced by γ radiation was significantly enhanced in MDA-MB-231 cells treated with 5–10 μm EGCG, accompanied by decreased NF-κB protein level and Akt phosphorylation [26]. These results raise the possibility that tea polyphenols may be a therapeutic adjuvant for human metastatic breast cancer. It is also noted that the polyphenolic compounds isolated from green tea can change the miRNA expression profile associated with angiogenesis in various cancer types [27,28]

In vitro experiments, phenolic compounds (PF) and (NPF), as contributing factors to the chemopreventive effects of green tea, both inhibit the migration of cancer cells by the disassembly of microtubules [29]. In a double-blind placebo-controlled study, oral administration of green tea catechin effectively inhibited cancer growth and reduced lower urinary tract symptoms, helping to treat symptoms of benign prostatic hyperplasia [30]. Moreover, another prior clinical trial experiment showed that daily intake of a standardized, catechin mixture containing 200 mg EGCG taken with food for 1 year was well tolerated and did not produce related side effects or adverse effects in men with HGPIN and/or atypical small acinar proliferation (ASAP) [31].

Resveratrol and trans-resveratrol (3,5,4′-trihydroxy-trans-stilbene), a phytoalexin discovered in more than 70 plant species like *Polygonum cuspidatum*, red grapes, berries, peanuts, and pines can be produced in response to mechanical injuries, ultraviolet radiation, as well as a defense for fungal infections [32,33]. Increasing studies reported that resveratrol possesses anticancer effects against several different tumor types by affecting multiple stages of tumor initiation and proliferation. Oral intake and intravenous administration are two routine administration methods of resveratrol, and oral intake is the major route. The enterocytes and hepatocytes are major metabolizing cells for resveratrol after oral administration, thus, resveratrol was often used to treat gastrointestinal and liver cancers. Intranasal administration of resveratrol at 105 mM for 25 weeks to A/J mice having 4-[methyl(nitroso)amino]-1-(3-pyridinyl)-1-butanone-induced lung carcinogenesis, showed a 27% decrease in tumor multiplicity and resulting in 45% decrease in tumor volume/mouse. While, oral administration of resveratrol failed to protect mice from chemically induced lung carcinogenesis [34]. Moreover, dietary suppression of dextran sulfate sodium (DSS)-associated tumorigenesis using resveratrol (300 ppm) supplementation for one week before the treatment was markedly exhibited in BALB/c background wild-type mice but not in Mkp-1−/− mice on the C57BL/6 background [35]. Additionally, the mitogen-activated protein kinase phosphatase 1 (Mkp-1) is required in the protective role of the Nrf2 signaling pathway against colitis-associated tumorigenesis. In addition, Kiskova et al. [36] reported that the combination of resveratrol and melatonin showed a reduction in *N*-methyl-*N*-nitrosourea (NMU)-induced mammary tumor incidence in SD rats by approximately 17% and decreased the quantity of invasive and in-situ carcinomas [36]. In addition, it is interesting to note that the chemopreventive effect of resveratrol (50, 100, and 300 mg/kg) against rat liver carcinogenesis was devoid of any adverse cardiovascular events. In addition, daily oral doses of resveratrol at 0.5 or 1.0 g tumor doses reduced the level of cell Ki-67 staining (a surrogate marker for cell growth) in the human gastrointestinal tract, by orders of magnitude enough to cause an anticancer effect [37,38]. The major metabolites of resveratrol were identified as sulfate and glucuronic acid conjugates of the phenolic groups and hydrogenation of the aliphatic double bond, such as trans-resveratrol (3,5,4′-trihydroxystilbene), trans-resveratrol 3-*O*-d-sulfate, and trans-resveratrol 4′-*O*-d-glucuronide [39,40]. Furthermore, the in vitro experiments also showed that the chemopreventive effect of resveratrol could be mediated by its metabolites [41].

Curcumin (bis-α, β-unsaturated β-diketone) is the most abundant polyphenol isolated from the roots of the perennial *Curcuma Longa* (Curcuma). Curcumin has been widely used as a remedy for many illnesses in different cultures [42,43]. In recent decades, increasing initial clinical studies focused on identifying whether curcumin plays an essential role in colorectal cancer models, due to its preferential distribution in the colonic mucosa. In a transgenic mouse model, phytosomal curcumin diets (150 mg curcuminoids/kg) exhibited significantly greater effects on inhibition of hepatocellular carcinoma formation, improvement of liver histopathology, and reduction of total tumor volume in transgenic mice, compared with unformulated curcumin [44]. Studies have shown that curcumin can be used as a chemoprophylaxis for ovarian cancer, and if taken daily, it can significantly and dose-dependently reduce the spontaneous incidence and tumor growth of ovarian cancer [45]. In addition, curcumin also showed an inhibitory effect in lung carcinogenesis induced by B[a]P, a procarcinogen present in the environment and cigarette smoke [46].

The study found that curcumin, a similar substance to A2, has the potential to be developed as a novel antiangiogenic drug. Curcumin can inhibit HGF- and promote EMT and angiogenesis by targeting c-Met and blocking PI3K/Akt/mTOR pathway. Curcumin analogs A2 can induce endothelial cell death mainly by enhancing NADH/NADPH oxidase-derived ROS [47,48]. In addition, combined exposure to bile with curcumin (50 and 100 μM) exhibited an inhibitory effect on acidic bile-induced oncogenic mRNA phenotype in a human hypopharyngeal primary cell (HHPC), including NF-ΚB, c-REL, compared to HHPC exposed to acidic bile without curcumin [49]. After treatment with 50 μM of curcumin, the invasion rate of thyroid cells through matrigel was decreased to 44.05 ± 4.64%, and scarcely any cells had migrated into the wound area during 24 h [21].

Following the intravenous injection of 20 mg/kg curcumin to mice, two metabolites of dihydrocurcumin (DHC) and tetrahydrocurcumin (THC) were detected in plasma samples. Additionally, it was distributed into liver, kidney and brain quickly with t_1/2z_ of 32.3 ± 10.8 min and AUC_0-∞_ of 107.0 ± 18.3 mg·min/L [50]. In previous human and animal studies, curcumin has been reported to undergo metabolic reduction through first-pass hepatic metabolism, which resulted in poor oral bioavailability and limited development of curcumin when orally administered. For this, a curcumin gum formulation alternating chewing and parking gum against buccal mucosa for 30 min, tested in healthy adult volunteers, showed higher release (1.67 g vs. 0.67 g) and absorption than chewing gum for 30 min, indicating a combination of chewing and parking the gum for prolonged mucosal contact could produce better release and absorption of curcumin [51].

#### 5.2.2. Flavones

Quercetin, a dietary bioflavonoid, that naturally occurs either as glycoside or as aglycone, is pharmacologically active and widely distributed in apples, onions, tomatoes, broccoli, and citrus fruits [52]. It can be considered as a potential chemopreventive compound due to its cardinal role in influencing the hallmarks of cancer as well as tumor-associated signaling pathways. A study by Zhang et al. [53] found that quercetin can induce tumor cell apoptosis by regulating the NF-B signaling pathway and its target genes Bcl-2 and Bax, suggesting that quercetin may be a candidate gene for oral squamous cell carcinoma (OSCC) chemoprevention [53]. Additionally, no tumor incidence was observed in SD rats treated with quercetin (200 mg/kg, o.p.), while the prostatic intraepithelial neoplasia (PIN) of 40% in the ventral prostate lobe and 20% adenocarcinoma was found in rats induced by hormone (testosterone) and carcinogen (MNU), respectively. The results showed that dietary quercetin could prevent the occurrence of ventral and dorsolateral prostate cancer induced by MNU + T in Sprague-Dawley rats [54]. Furthermore, an in vitro study showed that quercetin dosing of 50 μM decreased migration rates (26.36% vs. 64.36%) and motility rate by approximately tenfold in SK-MEL-28 cells cultured on collagen I matrices [55]. Additionally, another study suggested that quercetin can also be employed as an analgesic agent for cancer-related pain. They reported that when tested on Ehrlich tumor-induced pain, quercetin treatment (10–100 mg/kg, i.p.) has been shown to reduce mechanical and thermal hyperalgesia, but not affect tumor growth by decreasing hyperalgesia cytokines production, neutrophil recruitment, as well as activating opioid-dependent analgesic pathway [56].

Importantly, quercetin also enhanced the tumor growing inhibitory effect of some chemopreventive drugs, such as green tea, by improving their bioavailability. For instance, a follow-up study demonstrated that combination treatment of quercetin and green tea diet suggested a stronger inhibition of tumor growth in tumor-inoculated mice than green tea alone after 4 weeks of tumor inoculation, and inhibited the tumor growth by 47% compared to control. This cotreatment not only increased the tissue concentrations of total green tea polyphenols by 1.5-fold, but also increased nonmethylated EGCG by 1.8-fold [57,58].

Previous animal and human studies have shown that quercetin has poor bioavailability after a single oral administration because its absorption is affected by macronutrients. However, quercetin has a wide range of intestinal first-pass metabolism, which is absorbed by the intestinal tract and then enters phase II metabolism, and modulates the intestinal microbiota composition and protect the intestinal barrier. [59,60]. When 10 mg/kg of quercetin was orally administrated to rats, Chen et al. [61] found that about 93.3% of quercetin was metabolized in the gut, with only 3.1% metabolized in the liver, and the main forms of quercetin excreted in the rat bile were its glucuronide/sulfate and methylate conjugates [61]. Solid lipid nanoparticles (SLNs), a type of submicron particulate drug delivery system, may be available to enhance the absorption of poorly water-soluble drugs. The C_max_ value of quercetin in SLNs in a rat’s intestine was increased from 5.90 to 12.22 μg/mL, while the AUC (_0→48h_) was increased from 56.73 to 324.18 (μg/mL) × h [62].

Carotenoids belong to the chemical group of isoprenoid polyenes, which are natural fat-soluble pigments that provide bright coloration to plants and animals. Anabolism regulation of carotenoids in plants is a complex process, which is regulated by multiple factors [63]. Nowadays, there are many kinds of carotenoids, more than 750 kinds have been found in nature [64]. Carotenoids are commonly divided into four major classes: (1) vitamin A precursors like α, β-carotene; (2) pigments with partial vitamin A activity like cryptoxanthin; (3) non-vitamin A precursors like violaxanthin and neoxanthin; (4) non-vitamin A precursors like lutein and zeaxanthin. Recently, carotenoids have been found to also play an important role in human health; carotenoids dietary intake and serum and tissue levels of carotenoids are correlated with a lower risk of several types of cancer. Most noticeably, α-carotene had higher activity than β-carotene to suppress the tumorigenesis in the skin, lung, liver, and colon. Further evidence suggests that beta-carotene is more effective than retinoic acid (RA) in preventing hepatocellular carcinoma induced by diethylnitrosamine (DEN) and promoted by phenobarbital (PB) [65].

Another carotenoid compound of lycopene was originally developed as a cancer chemopreventive agent for prostate cancer. One of the studies indicated that lycopene supplementation (at doses of 15 mg/day) for 6 months in elderly men diagnosed with benign prostate hyperplasia (BPH) inhibited serum prostate-specific antigen (PSA) increase, and further improving clinical diagnostic markers and symptoms of BPH [66]. Furthermore, the major metabolite of lycopene, Apo-10′-lycopene acid, in the treatment of human liver The-2 and Huh7 cells, can upregulate sirtuin 1 (SIRT1) by stimulating the SIRT1 signaling pathway, which is a NAD(+)-dependent protein deacetylase and dose-dependent inhibition of cell growth. Dietary supplementation of 10 mg/kg Apo10La for 24 weeks significantly reduced the ability of the DEN-initiated high-fat diet (HFD) to promote liver tumors (tumor diversity decreased by 50%; 65% volume) and the incidence of lung tumors in C57BL/6J mice (reduced by 85%), which also improved glucose intolerance and reduced liver inflammation in HFD mice, and was considered an as an effective chemopreventative agent against hepatic tumorigenesis and inflammation [67]. Similarly, epidemiological studies have shown that dietary supplementation with lutein and zeaxanthin is associated with a lower risk of various cancers. Studies have shown that zeaxanthin can induce apoptosis of human uveal melanoma cells in vitro, or inhibit the growth of various tumor cell lines. Lutein can inhibit the PI3K/Akt signaling pathway, induce the apoptosis of lung cancer cells (A549) and reduce the risk of lung cancer without side effects, all of which may be effective natural anticancer drugs [68,69].

Carotenoids are mostly fat-soluble, following the same intestinal absorption path as dietary fat, and therefore, the absence of bile or any generalized malfunction of the lipid absorption system will interfere with the absorption of carotenoids [70]. The carotenoid bioavailability from foodstuffs may be largely affected by the food matrix and other dietary components. It has been reported that carotenoid absorption may be the greatest when daily recommended vegetables are consumed in one meal compared to smaller doses over multiple meals [71].

#### 5.2.3. Monoterpene and Triterpenoids

Over the last two decades, extensive research on plant-based medicinal compounds has revealed a potential role of triterpenoids in prevention. Thymoquinone (2-methyl-5-isopropyl-1,4-benzoquinone) is a monoterpene derived from the seed oil of *Nigella sativa* L. (family Ranunculaceae), first isolated from black seed in 1963 by El-Dakhakhany [72]. A large number of studies in vivo and in vitro have indicated that thymoquinone suppressed experimentally carcinogenesis in the colon, breast, and skin, and its mechanism may be associated with suppression on the AKT and ERK signaling pathways. Several studies reported that thymoquinone supplementation in drinking water resulted in significant suppression on forestomach, lung cancer, and hepatic carcinogenesis induced by benzo(α) pyrene, 20-methyclonathrene diethylnitrosamine (DENA), and diethylnitrosamine, respectively [73,74,75]. Additionally, thymoquinone at various concentrations ranging from 50 to 100 nM effectively inhibited migration, invasion, and tube formation of human umbilical vein endothelial cells (HUVEC). This study also suggested when given by subcutaneous injection, thymoquinone (6 mg/kg) inhibited human prostate tumor growth in both size and weight in a xenograft human prostate cancer (PC3) model in mice, and this low dosage almost has no chemotoxic side effects [74]. Oral administration of thymoquinone had a slower absorption characteristic as compared with elimination, but with good bioavailability, with a T_1/2_ value of 274.61 min and C_max_ of 3.48 μg/mL [76]. In addition, geraniol, an acyclic monoterpene, represents a potential chemopreventive agent in colon carcinogenesis, as indicated by the decreased number of total aberrant crypt foci (ACF) and ACF ≥ 4 crypts in the distal colon [77]. In in vivo and in vitro experiments, geraniol can also induce PC-3 prostate cancer cells to inhibit the growth of prostate cancer by targeting cell cycle and apoptosis pathways in cultured cells and tumor grafted mice and regulate the expression of various cell cycle regulators and Bcl-2 family proteins in PC-3 cells [78].

Triterpenoids, such as betulinic acid, lupeol, oleanolic acid, and cucurbitacin, are metabolites of isopentenyl pyrophosphate oligomers and are ubiquitously distributed in the form of free triterpenoids, triterpenic glycosides (saponins), and their precursors with potential cancer prevention properties [79,80].

In a colon cancer model, doses of oleanolic acid (OA, 3b-hydroxyolean-12-en-28-oic acid) ranging from 250 to 1500 ppm inhibited mean colonic aberrant crypt foci (36–52%) induced by azoxymethane (AOM) in a dose-dependent manner in male F344 rats [81]. Daily intraperitoneal injections of oleanolic acid (12.5 mg/kg, i.p.) for 16 days have been found to inhibit tumor growth and reduce intratumoral microvessel density (MVD) in xenograft tumor tissues of colorectal cancer mice [82]. However, the systemic absorption of oleanolic acid was extremely low for oral doses of 25 and 50 mg/kg in rats, that may be due to poor gastrointestinal absorption and subsequent hepatic first-pass metabolism [83]. Therefore, other direct blood delivery methods, such as intraperitoneal injections, are more suitable for ursolic acid on cancer chemoprevention.

The chemoprophylaxis of 2-cyano3, 12-dioxane-1, 9 (11)-diene-28-OIC acid (CDDO) against tumor growth in xenograft mouse models of breast cancer was studied in vivo and in vitro. It not only effectively inhibits the activity of HER2 tyrosine kinase, but also inhibits the growth of HER2 overexpressed breast cancer cells [84]. Moreover, methyl ester or ethyl amide derivatives of CDDO showed more potent for delaying the initial development of tumors, than the individual drug of suberoylanilide hydroxamic acid (SAHA), with a 7-week delay before 50% tumor incidence was reached [85].

Another triterpenoid compound of betulinic acid has also been reported to possess antiangiogenic effects by inhibiting aminopeptidase N, an enzyme that is involved in the regulation of angiogenesis induced by growth factor in endothelial cells, possibly by affecting mitochondrial functions [86,87]. Compared to betulinic acid, 20, 29-dihydro-betulinic acid derivatives were found to exert better antiangiogenic properties than betulinic acid [88]. Betulinic acid is confirmed as an agent with rapid absorption and slow biphasic disappearance from serum. After a single dose of 500 mg/kg intraperitoneal betulinic acid, the serum concentrations reached peaks at 0.23 h, and the elimination half-lives were detected as 11.8 h. The distribution of betulinic acid in tissues at 24 h in descending order was as follows: perirenal fat, ovary, spleen, mammary gland, uterus, and bladder [89].

#### 5.2.4. Sulfur Compounds

As we all know, the anticarcinogenic effect of *Allium* vegetables including garlic is attributed to organosulfur compounds, such as allicin, which exhibited a high protective effect against cancer in animal models induced by a variety of chemical carcinogens [90]. Allicin is responsible for the characteristic odor of garlic and has been shown to have various health-promoting effects, including cancer chemopreventive actions. It is reported that allicin inhibited lymphangiogenesis suppressing activation of vascular endothelial growth factor (VEGF) receptor, which is a critical cellular process implicated in tumor metastasis [91]. Additionally, allicin also can be served as adjunctive therapy for thyroid cancer, as it induces autophagic cell death to alleviate the malignant development of cancer [92]. Allicin and other thiosulfinates decompose instantly to oil-soluble organosulfur compounds in a short of time, including diallyl sulfide (DAS), diallyl disulfide (DADS), diallyl trisulfide (DAT), and diallyl tetrasulfide (DATS). A study by Lai et al. documented that DAS, DADS, and DATS may become the potential of the application of drug resistance, including DATS to colo 205 cells migrating and invading the strongest inhibitory effect [93]. Furthermore, administration of DAS (10 mg/kg, i.p.) also significantly reduced the number of tumor-directed capillaries (16.5 vs. 28.75) in male C57BL/6 mice, compared to the control group, through enhancing the production of antiangiogenic factors such as IL-2 and TIMP [94]. Additionally, the regulating effect on the metabolism of the carcinogen is considered as one of the possible mechanisms for protection against cancers of DAS [95].

Sulforaphane is the most characterized isothiocyanate (ITC) compound, mainly found in high concentrations in broccoli, which is viewed as a conceptually promising agent in breast cancer prevention. It can be rapidly absorbed and achieved a peak level before 1 h [96]. Sulforaphane can inhibit the translocation of the NF-κB p65 subunit, downregulate p52 and its downstream transcriptional activity, and preferentially eliminate breast cancer stem cells (CSC). Sulforaphane combined with docetaxel significantly reduces primary tumor volume and secondary tumor formation compared to treatment alone [97]. However, sulforaphane is superior to glucoraphanin in modulating phase I and phase II enzymes involved in carcinogen metabolizing in vitro [98]. Furthermore, whether sulforaphane exerts a direct chemopreventive action on animal and human mammary tissue was determined. Following with oral dosing of a broccoli sprouts preparation containing 200 μmol of sulforaphane, sulforaphane metabolites of dithiocarbamate were readily measurable in human breast tissue enriched for epithelial cells, providing a strong rationale for evaluating the protective effects of sulforaphane in clinical trials of women at risk for breast cancer [99]. In addition, orally ingested other ITC compounds, such as allyl isothiocyanate (AITC), benzyl isothiocyanate (BITC), erucin (ECN), phenethyl isothiocyanate (PEITC), and sulforaphane (SF), were selectively delivered to the bladder via urinary excretion, and possess promising chemopreventive activities against human bladder cancers [100].

#### 5.2.5. Cellulose

Micronutrients, including vitamin D, vitamin E, folic acid, calcium, and selenium, are currently being evaluated in National Cancer Institute (NCI)-sponsored chemoprevention trials for prostate, colon, and breast cancers.

Selenium is considered as one of the most promising agents currently under laboratory development for prostate cancer chemoprevention. Low selenium levels are associated with a higher risk of prostate, lung, and colon cancer. Selenium may have a protective effect on cancer, and individuals with selenium deficiency may benefit from supplemental selenium, thereby reducing their risk of cancer [101]. In addition, compelling evidence that selenium-induced growth inhibition and apoptosis in PC-3 prostate cancer cells were found to be dose-dependent [102]. However, it is worth noting that Se compounds can be either cytotoxic or possibly carcinogenic at higher concentrations, which is indicated to be associated with oxidative stress [103].

Calcium is not only an essential structural component of the human body but also a key functional element for maintaining cellular structure. Calcium is considered to be an effective chemopreventive agent, and increased intracellular Ca^2+^ concentration enables cytotoxic T lymphocytes (CTL) and natural killer (NK) cells to achieve effective functional expression, thus killing cancer cells [104]. In an in vivo investigation, calcium inhibited colonic epithelial cell proliferation induced by heme in rats, which suggested that calcium might decrease the colon cancer risk related to a high intake of red meat [105]. There have also been studies supporting the use of intravenous CA/MG as an effective neuroprotective against oxaliplatin-induced cumulative sNT in adjuvant colon cancer [106]. There is also some experimental evidence that calcium can protect against breast cancer development, However, calcium intake was not associated with breast cancer risk in the Singapore Chinese Health Study, comparing highest quartile to lowest quartile of intake [107].

1,25-Dihydroxy vitamin D, a naturally occurring, biologically active form of vitamin D, is an important modulator for the absorption and metabolism of calcium, which can become deficient as a result from inappropriate diet or inadequate exposure to sunlight. Therefore, vitamin D also has a role in preventing the malignant transformation and the progression of various types of human tumors while modulating calcium metabolism. The prophylactic effects of vitamin D on the most common types of cancer have been extensively investigated either in vitro or in vivo. The inhibitory effect of vitamin D on tumor cell growth was first described by Colston et al. The reported that decreased cell proliferation was found in melanoma cells treated with 1,25-dihydroxyvitamin D3 (1,25(OH)_2_D_3_) [108]. Subsequently, the growth suppressing the activity of 1,25(OH)_2_D_3_ was observed in other tumor cell lines such as colon, lung, and prostate cancer cells [109,110]. These studies also highlighted the presence of a specific vitamin D receptor (VDR) that appeared to be essential for the growth inhibitory activity of 1,25(OH)_2_D_3_. In line with these investigations, other in vitro studies revealed that decreased intracellular levels of VDR markedly reduced the sensitivity of 1,25(OH)_2_D_3_ towards the antiproliferative effects [111]. The major obstacle of 1,25(OH)_2_D_3_ in the clinical development was the reason of induced hypercalcemia at pharmacologically relevant doses [112]. In addition, some major studies also exhibited that non-alpha-tocopherol forms of vitamin E hold considerable promise for the chemoprevention of prostate cancer [113].

## 6. Possible Mechanisms of Inhibition of Carcinogenesis

### 6.1. Induction of Apoptosis

Dysregulation of apoptosis can lead to tumorigenesis, causing the production of a malignant phenotype, rendering resistance to chemotherapy and radiotherapy [114]. Therefore, stimulating apoptosis in premalignant and malignant cells is regarded as an important means of chemoprevention. Curcumin was reported to play a time and dose-dependent induction of apoptosis in the human colon cancer HCT116 cells, which is related to the activation of c-jun N-terminal kinase (JNK) and caspase 3, and the inhibition of nuclear factor-kappa B (NF-κB) transcription activity. Curcumin treatment can also induce apoptosis by inducing JNK-dependent c-jun continuous phosphorylation and stimulating AP-1 transcriptional activity [115]. Curcumin can reduce hypoxia-induced oxidative stress and inflammatory cytokines, improve cell survival, and play a pro-apoptotic role by enhancing the expression of peroxidase 6 (PRDX6) and inhibiting the release of NF-κB to achieve the purpose of chemoprophylaxis [116]. ECGC can enhance cancer cell apoptosis induced by endoplasmic reticulum stress by inhibiting PARP activity [117]. ECGC can downregulate the antiapoptotic factors BCL-2, BCL-XL and XIAP and upregulate the proapoptotic factors BAD and BAX in human adrenal cancer NCI-H295 cells [118]. EGCG induces apoptosis through inhibiting the phosphorylation of Akt in the PI3K-Akt signaling pathway in human pancreatic cancer PANC-1 cells, upregulating the expression of Caspase-3 and Bax, and downregulating the expression of Bcl-2 [119]. Resveratrol inhibits prostate cancer by reducing cell proliferation and migration and inducing apoptosis through the Akt/microRNA-21 pathway [120]. A study showed that resveratrol also induced apoptosis in L-428 cells of Hodgkin’s lymphoma (HL) origin when treated at a higher concentration range (50 μm) for 48 h [121]. [10]-gingerol treatment induced the increase of the Bax/Bcl-2 ratio and activated caspase-9 and caspase-3 in a dose-dependent manner, which means induction of apoptosis [122]. The above-mentioned related potential mechanisms are shown in Figure 1.

### 6.2. Inhibition of Cell Signaling Associated with Migration and Proliferation

Matrix metalloproteinases (MMPs) are an important prerequisite for tumor invasion and metastasis [123]. After the treatment of prostate cancer DU-145 cells with curcumin, the expression of MMP-2 and MMP-9 was significantly reduced, and the in vitro invasion ability was inhibited. Curcumin has been shown to significantly reduce tumor volume, and activity of MMP-2 and MMP-9 at the tumor-bearing site [124,125]. MMP-9 and VEGF are involved in tumor cell invasion and metastasis [126], and curcumin-mediated inhibition of human tongue squamous carcinoma CAL-27 cells invasion may be partly attributed to the downregulation of NF-kB and its downstream target genes such as MMP-9 and VEGF [127]. An experimental study showed that resveratrol can inhibit MMP-9 expression by upregulating peroxisome proliferators-activated receptors (PPAR) α [128]. Resveratrol may also downregulate the expression of MMP-2 in hepatoma in nude mice through the NF-κB pathway to inhibit tumor growth and invasion ability [129]. ECGC also inhibits the activity of MMP-9 [130] and MMP-2 [131] to inhibit cell migration. Cell cycle arrest is closely related to cell growth, which is closely linked to cell proliferation. In addition, curcumin has an effective sensitization effect and can induce G1 cell cycle arrest of colon cancer HCT116 cells through cyclin-dependent kinase 2 (CDK2) as a direct target, effectively inhibiting the proliferation of HCT116 cells [132]. Similarly, curcumin can be dose-dependent. After treatment with 40 μmol/L curcumin for 24 h, autophagy induces apoptosis and upregulates the expression of NF-κB, which significantly inhibits the proliferation of human gastric cancer cells (SGC7901) [133]. Resveratrol can downregulate the expression of human telomerase reverse transcriptase (hTERT) protein in a concentration-dependent manner and inhibit the ability of telomerase, which is an important mechanism of resveratrol on the proliferation of human epidermoid carcinoma (A431) cells [134]. 6-gingerol suppressed cyclin D1 expression [135], and EGCG has an antiproliferative effect on colorectal cancer cells through the degradation of cyclin D1 and the upregulation of p21 [136]. The above-mentioned related potential mechanisms are shown in Figure 2.

## 7. Conclusions and Future Prospects

This review mainly summarizes five types of natural compounds that can play a chemopreventive effect, namely polyphenols, flavonoids, monoterpene and triterpenoids, sulfur compounds, and cellulose. Contents of the main plant sources, dosages, metabolites, chemopreventive effects, etc., are introduced. These natural compounds play a chemopreventive role through the regulation of apoptosis, migration, proliferation, and other mechanisms ultimately. However, most of the chemopreventive effects of natural products are limited to basic research on cells and animals, only a few natural products such as resveratrol and EGCG have undergone clinical research. Moreover, the corresponding toxicological research is relatively lacking, and the mechanism research is still incomplete.

The continuous increase in the incidence and mortality of cancer has become one of the main factors threatening the health and life safety of human beings. Although there have been new breakthroughs in the treatment of cancer in recent years, the treatment of most cancers, especially the advanced cancers, is still difficult. The occurrence and development of cancer is multistep, multipath, and multifocal. Cancer chemoprevention refers to the application of natural products or synthetic chemicals to prevent, slow down, or even reverse the development of cancer. Compounds isolated from plants can target various events in the carcinogenic process. Increasing studies have shown that they could serve as promising resources for cancer prevention and therapeutic intervention. The evidence reported in this review highlights the continuous development and innovative application of natural products, indicating that it occupies an increasingly critical position in the source of anticancer compounds. The development and application of anticancer drugs have gradually shifted from drugs with a single target and strong side effects to natural drugs with less toxicity that are regulated by multichannel and multitarget. Each natural medicine usually affects more than one pathway (such as triggering apoptosis, inhibiting cell proliferation, etc.). The multitarget property of natural medicines enables them to effectively offset the biological complexity of cancer, which shows that plant natural products offer promising resources for cancer chemoprevention.

Therefore, it is necessary to increase the mechanism research and clinical research of the natural products in future research, and to improve its toxicology research to conduct a more comprehensive study on the chemopreventive effect of natural products.

## Figures and Tables

**Figure 1 molecules-26-00933-f001:**
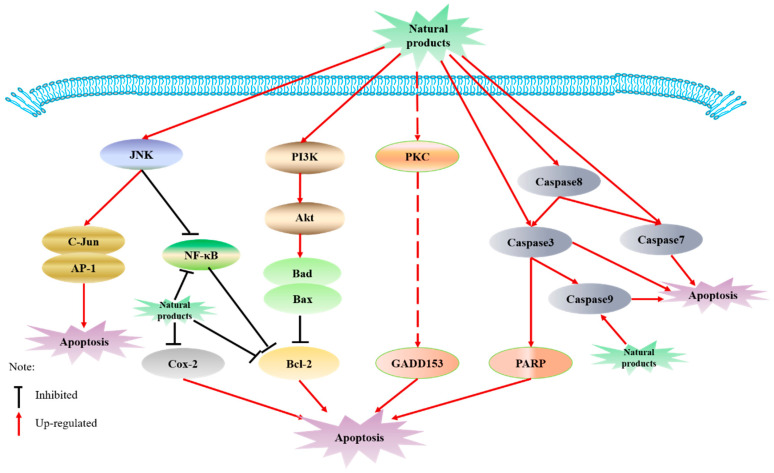
Mechanism of apoptosis.

**Figure 2 molecules-26-00933-f002:**
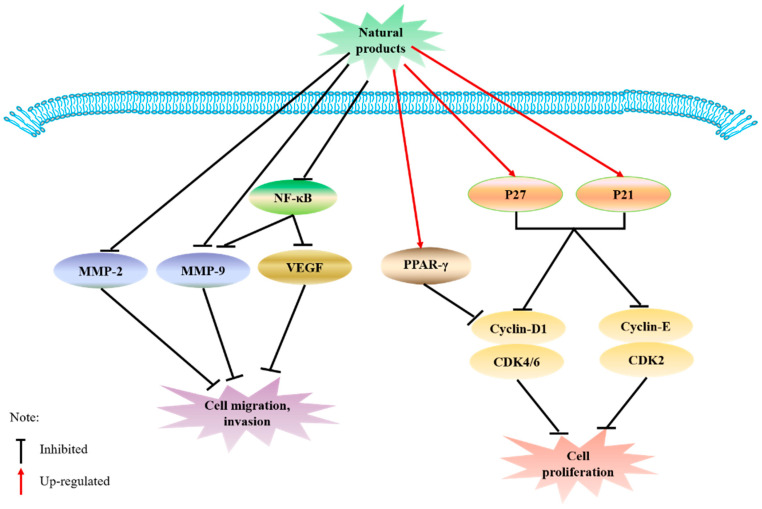
Mechanism of migration and proliferation.

**Table 1 molecules-26-00933-t001:** Chemopreventive properties of phytochemicals.

Common Phytochemicals
Groups	Phytochemicals	Structure	Cancer Type	The Subjects	Chemopreventive Property	Common Source	Reference
Phenols	(−)-epicatechin (EC)	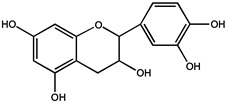	Prostate cancer	Human	Oral administration of 600 mg/d for 1 year reduced the incidence of diagnosed cancers in volunteers with high-grade intraepithelial neoplasia of the prostate (H GPIN)	Green tea	[30]
(−)-epigalocatachin (EGC)	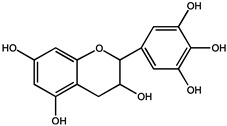	Breast cancer	MCF10A cells	Completely inhibited Met, AKT and ERK phosphorylation at 0.6 mM	[27]
(−)-epicatechin-3-gallate (ECG)	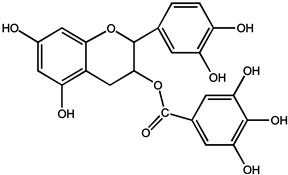	Liver cancer	HepG2 cells	Interrupted closures by the disassembly of microtubules	[28]
Osteosarcoma cancer	U2OS cells
(−)-epigallocatechin-3-gallate (EGCG)	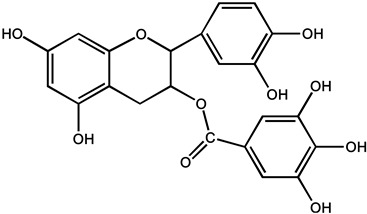	Hepatocellular carcinoma	HuH7 cells	Inhibited the growth of HuH7 xenografts	[25]
Breast cancer	Tumorigenic breast epithelial cells	Blocked the ability of hepatocyte growth factor (HGF) to induce cell motility and invasion	[27]
Prostate cancer	Human	Not produce treatment related adverse effects in men with baseline HGPIN or ASAP	[31]
Resveratrol	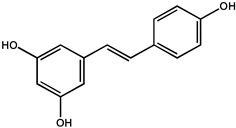	Lung cancer	A/J mice	Tumor diversity and volume decreased in mice	*Polygonum cuspidatum,* red grapes, berries, peanuts, pines etc.	[34]
Colorectal cancer	BALB/c wild-type mice	Marked suppression of dextran sulfate sodium (DSS)-associated tumorigenesis	[35]
Human	It can be used as a potential chemoprophylaxis for colorectal cancer tract	[37]
Curcumin	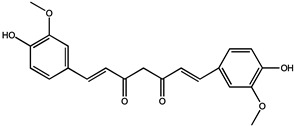	Liver cancer	Transgenic mice	Inhibition of hepatocellular carcinoma formation, improvement of liver histopathology, and reduction of total tumor volume in transgenic mice		[44]
Ovarian cancer	Hens	Reduced the overall ovarian cancer incidence to 31% and 57%		[45]
Lung cancer	Lung cancer cells (H1299, A549)	An inhibitory effect in lung carcinogenesis induced by B[a]P, a procarcinogen present in environment and cigarette smoke	Curcuma Longa	[46]
Curcuma Longa A2	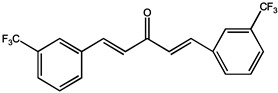	ROS-dependent endothelial cell	Human umbilical vein endothelial cells (HUVECs)	Suppresses the migration and tube formation of human umbilical vein endothelial cells (HUVECs) in vitro	[47]
——————	Rat	Suppresses newly formed microvessels in chicken chorioallantoic membranes (CAMs) and Matrigel plus in vivo
Flavonoids	Quercetin	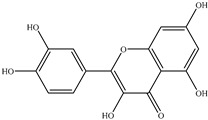	Oral squamous cell carcinoma (OSCC)	Hamster	Decreased incidence of oral squamous cell carcinoma (OSCC) and severity of hyperplasia and dysplasia	Apples, onions, tomatoes, broccoli, citrus fruit, etc.	[34]
Melanoma	SK-MEL-28 human melanoma cells	Decreased migration rates (26.36% vs. 64.36%) and motility rate by approximately tenfold in SK-MEL-28 cells cultured on collagen I matrices	[55]
Lutein	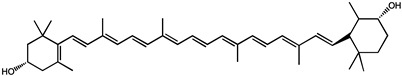	Lung and colon cancers	Human	Reduce the risk of lung and colon cancers by the suppression of k-Ras and β-catenin expression	Papaya, pumpkin, citrus, wolfberry, peach, spinach, leek, corn, Chinese cabbage, etc.	[68]
Zeaxanthin	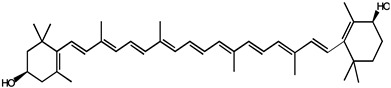
Lycopene	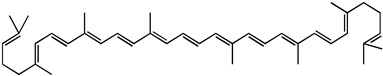	Benign prostate hyperplasia (BPH)	Human	Inhibited serum prostate-specific antigen (PSA) increase, and further improving clinical diagnostic markers and symptoms of BPH	Tomato, tomato products, watermelon, grapefruit etc.	[66]
	Apo-10-lycopenoic	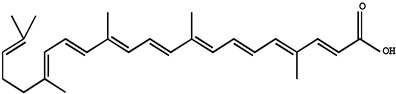	Liver tumors	HFD-fed mice	APO10LA can effectively inhibit HFD-promoted hepatic tumorigenesis by stimulating SIRT1 signaling while reducing hepatic inflammation	[67]
Monoterpene	Thymoquinone	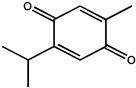	Human prostate cancer	PC3 cancer cells	Inhibits tumor angiogenesis and tumor growth and could be used as a potential drug candidate for cancer therapy	The seed oil of *Nigella sativa* L.	[74]
——————	Human umbilical vein endothelial cell (HUVEC)	Effectively inhibited migration, invasion, and tube formation of human umbilical vein endothelial cell (HUVEC)	[74]
Human prostate cancer (PC3)	Male mice	Inhibited human prostate tumor growth in both size and weight in a xenograft human prostate cancer (PC3) model in mice	[74]
Triterpenoids	Oleanolic acid	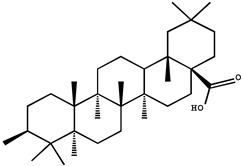	Colon cancer	Male mice	Oleanolic acid inhibited, in a dose-dependent manner, the average azomethane (AOM)-induced abnormal colonic cavitation lesions in male F344 rats (36–52%)	Hedyotis Herbaherba, hawthorn, Syzygium Aromaticum, loquat leaf etc.	[81]
Colorectal cancer	Mice	Inhibitory tumor growth of xenograft tumor tissue in mice with colorectal cancer	[82]
2-cyano-3,12-dioxooleana-1,9(11)-dien-28-oic acid	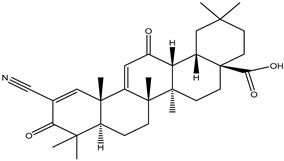	Breast cancer	Immunodeficient mice	CDDO (20 mg/kg, i.v.) treatment for 3 weeks abrogated the growth of both MCF7/HER2 and MDA-MB-435/HER2 tumors types in immunodeficient mice, by inhibiting HER2 phosphorylation and decreasing HER2 kinase activity	[84]
Betulinic acid	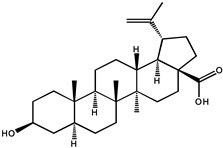	Colorectal cancer	Nude mice	Possess antiangiogenic effects by inhibiting aminopeptidase N	White birch bark, Ziziphi Spinosae, semen	[87]
Sulfur compounds	Allicin	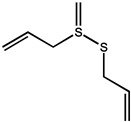	——————	——————	Inhibited lymphangiogenesis suppressing activation of vascular endothelial growth factor (VEGF) receptor	Garlic, allium, vegetables etc.	[91]
Thyroid cancer	SW1736 and HTh-7 cells	Served as an adjunctive therapy for thyroid cancer, as it induces autophagic cell death to alleviate the malignant development of cancer	[92]
Diallyl sulfide	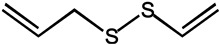	——————	C57BL/6 mice	Compared to the control group, through enhancing the production of antiangiogenic factors such as IL-2 and TIMP	[94]
Diallyl tetrasulfide	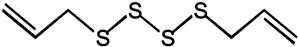	Colon cancer	CoLo 205 cells	Three oil-soluble compounds including DAS, DADS, and DAT at 10 and 25 μM have an inhibitory effect on the migration and invasion of human colon cancer cells with the order of DATS < DADS < DAS	[93]
Diallyl disulfide	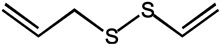
Diallyl trisulfide	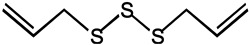
Sulforaphane	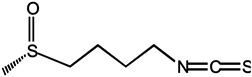	Breast cancer	Rats	As a conceptually promising agent in breast cancer prevention. It can be rapidly absorbed and achieved a peak level before 1 h	Broccoli	[96]
Triple negative breast cancer	Orthotopic mouse xenograft model	The addition of sulforaphane can prevent the expansion and clearance of breast CSCs, which will greatly benefit the treatment of TNBC with cytotoxic chemotherapy	[97]
Cellulose	Selenium	————	Prostate cancer	LNCaP cells	Selenium-induced growth inhibition and apoptosis in PC-3 prostate cancer cells were found to be dose dependent	————	[102]
Calcium	————	Colon cancer	Rats	Inhibited colonic epithelial cell proliferation induced by heme in rats, which suggested that calcium might decrease the colon cancer risk related to high intake of red meat	[105]
Colon cancer	Human	Intravenous CA/MG can be used as an effective neuroprotectant against the accumulation of SNT in adjuvant colon cancer induced by oxaliplatin	————	[106]
1, 25-Dihydroxy vitamin D3	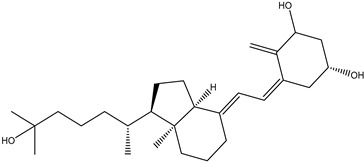	Melanoma tumor	Melanoma cells	Decreased cell proliferation was found in melanoma cells	————	[108]
Prostate cancer	Human	Slow the rate of prostate specific antigen (PSA) rise in PCa patients demonstrating proof of concept that 1,25(OH)2D3 exhibits therapeutic activity in men with PCa	[109]

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
