# Peer review of "Plant Natural Products: Promising Resources for Cancer Chemoprevention"

_molecules, 2021, doi:10.3390/molecules26040933_

Round 1

Reviewer 1 Report

In a situation where cancer is the second highest cause of death, it is very important to prevent cancer and try appropriate treatment at each stage of cancer occurrence. Although various treatment methods exist, discovering substances that are effective in preventing and treating cancer, especially from natural products, has many advantages. This review well summarizes five types of natural compounds that can play an chemopreventive effect, namely polyphenols, flavonoids, monoterpene and triterpenoids, sulfur compounds and cellulose. In particular, it explains well how each natural product causes induction of apoptosis in premalignant and malignant cells. This review is considered to have sufficient qualifications for publication because it has well organized the role of natural products in cancer prevention and treatment, especially in prevention.

Author Response

First of all, thank you for taking time out of your busy schedule to review the manuscript and your positive comments, which greatly encouraged me and enabled me to make a better summary in the revision of the manuscript with greater confidence. With the trend of the younger and younger age of cancer onset, I am more aware of the importance of effective substances for the prevention and treatment of cancer. A large number of in vivo and in vitro experiments have shown that the chemoprophylaxis of natural products is not only extensive but also common. Bioactive ingredients extracted from plants will provide valuable resources for the development of new anticancer drugs.

Thank you again for your positive guidance and I hope to learn more from you.

Best regards.

Reviewer 2 Report

The review entitled: “Plant Natural Products: Promising resources for cancer chemoprevention” is a very interesting and well organized review.

The structure of article confirms to principles required for review article. The authors have studied and used sufficient bibliography sources quoted in the article. It confirms the evidence of the theoretical knowledge and very good orientation within the problem discussed in the article. The words processing of the article is adequate. The article fulfils the formal requests on required level, together with clearly established aims and objectives. The article was written in legible clear and understandable to the reader. Methods used by the researchers were not very sophisticated.

It certainly can be published in Molecules after minor correction since it is too long and therefore not concise. The review contains a lot of information, but its perception is very limited, mainly due to the too detailed description. The most important data is contained in the tables, but the text is unnecessarily so expanded in those already presented data. It should be noted that the authors should present the most important results in a given scope of the topic, and not repeat detailed descriptions of experiments and results.

The authors should correct the reference list, as in some sections the cited works do not seem to be up-to-date for cancer research. Analyzing the “age” of the cited publications, it can be seen that 71 out of 140 cited publications are much older than 10 years, which is debatable in the rapidly developing field of research on anticancer properties of plant ingredients. It should be emphasized that good review work should contain the latest data from the last few years.

In conclusion I do recommend the manuscript for further steps of Molecules publication process, after the indicated revision.

I appreciate your kind invitation as the external reviewer of the valuable scientific topic.

Please feel free to contact me if you have any question or concern on the review report.

Author Response

First of all, thank you for taking time out of your busy schedule to review this manuscript. Your positive comments and suggestions have important guiding significance for my thesis writing and scientific research work. I have made further corrections to the manuscript by your instructions. The revised results are summarized as follows:

  1. As for the content of the article, I mainly cut some details of the experiment and extracted the most important results obtained in the experiment in view of various natural compounds in the fifth part of the article. According to the changes in the content of the article, the table has been modified accordingly, and most of the main data of some experiments have been described in the table. In order not to affect the logic of the overall framework of the article by a large numberof changes, the expression of some concise experimental details and data results is retained, to increase evidence on the one hand and make readers understand more intuitively on the other hand.
  2. Aboutthe revision of the references, I updated or replaced some of the chapters on the research contents of natural compounds for the prevention of cancer in the past few years. A small number of references are older but have been retained because of the first references to proper nouns involved. Some other references between 2006 and 2013, due to their high reference value and a large number of references, have been revised several times and think that their retention may make the content of the article more reliable.

Finally, thank you again for your valuable advice. You let me know how to use the latest data of the past few years to prove problems and strengthen my ability to write papers. I hope I can learn more knowledge from you in the future.

Best regards.